# The Case for Pre-Emptive Pharmacogenetic Screening in South Africa

**DOI:** 10.3390/jpm14010114

**Published:** 2024-01-19

**Authors:** Tracey Hurrell, Jerolen Naidoo, Collen Masimirembwa, Janine Scholefield

**Affiliations:** 1Bioengineering and Integrated Genomics Group, Future Production Chemicals Cluster, Council for Scientific and Industrial Research, Pretoria 0001, South Africa; thurrell@csir.co.za (T.H.); jnaidoo@csir.co.za (J.N.); 2African Institute of Biomedical Science and Technology, Harare 00263, Zimbabwe; cmasimirembwa@aibst.edu.zw; 3Sydney Brenner Institute for Molecular Biology, Division of Human Genetics, Faculty of Health Sciences, University of the Witwatersrand, Johannesburg 2193, South Africa; 4Department of Human Biology, Faculty of Health Sciences, University of Cape Town, Cape Town 7925, South Africa; 5Division of Human Genetics, Faculty of Health Sciences, University of the Witwatersrand, Johannesburg 2193, South Africa

**Keywords:** Africa, pharmacogenetics, adverse drug reactions, pre-emptive screening

## Abstract

Lack of equitable representation of global genetic diversity has hampered the implementation of genomic medicine in under-represented populations, including those on the African continent. Data from the multi-national Pre-emptive Pharmacogenomic Testing for Preventing Adverse Drug Reactions (PREPARE) study suggest that genotype guidance for prescriptions reduced the incidence of clinically relevant adverse drug reactions (ADRs) by 30%. In this study, hospital dispensary trends from a tertiary South African (SA) hospital (Steve Biko Academic Hospital; SBAH) were compared with the drugs monitored in the PREPARE study. Dispensary data on 29 drugs from the PREPARE study accounted for ~10% of total prescriptions and ~9% of the total expenditure at SBAH. VigiLyze data from the South African Health Products Regulatory Authority were interrogated for local ADRs related to these drugs; 27 were listed as being suspected, concomitant, or interacting in ADR reports. Furthermore, a comparison of pharmacogene allele frequencies between African and European populations was used to frame the potential impact of pre-emptive pharmacogenetic screening in SA. Enumerating the benefit of pre-emptive pharmacogenetic screening in SA will only be possible once we initiate its full application. However, regional genomic diversity, disease burden, and first-line treatment options could be harnessed to target stratified PGx today.

## 1. Introduction

Pharmacogenomics (PGx) investigates the pharmacological consequences of variations in genes relevant to the pharmacokinetics and pharmacodynamics of pharmaceuticals and has been a recognized science since the 1950s, with biochemical individuality being predicted decades prior. However, despite noteworthy disparities in drug responses across diverse genetic backgrounds, rooted as far back as primaquine during World War 2, the clinical implementation of pharmacogenomic-based healthcare has been complex [1]. The lack of equitable representation of global genetic diversity in resources such as the Human Genome Project and Genome-Wide Association Studies (GWAS) datasets has also hampered the implementation of genomic medicine advances in under-represented population groups, including those on the African continent [2]. The integration of genomics into healthcare has increasingly been adopted in developed countries over the past decade, with government investments across 14 countries (including the United Kingdom, Netherlands, Switzerland, China, Australia, and Denmark) exceeding USD 4 billion. In contrast, to date, there are no active government-funded national genomic medicine programs under way on the African continent, thereby limiting the realization of the benefits of genomic healthcare for the continent [3].

A major historical barrier to adopting pharmacogenomic testing for clinical therapeutic recommendations has been the lack of guidelines. Over the past two decades, several committees, including the Canadian Pharmacogenomics Network for Drug Safety [4], the Dutch Pharmacogenetics Working Group (DPWG) [5], and the Clinical Pharmacogenetics Implementation Consortium (CPIC) [6], have been established to develop clinical guidelines. These guidelines are based on several sources of evidence, including preclinical functional outcomes, clinical trial data, and updated consensus information. Over a ten-year period following its inception (2009–2020), the CPIC published 23 separate actionable guidelines spanning 46 drugs from various therapeutic areas, which included 19 genes [7]. Central to establishing guidelines is the Pharmacogenomics Knowledge Base (PharmGKB, https://www.pharmgkb.org/ accessed on 8 September 2023), which utilizes a clinical annotation scoring system to assign a level of evidence to support the association between drug–gene pairs [8]. While each committee’s guidelines are nuanced, their evidence-based recommendations support clinical decision making [9]. However, further co-ordination between regulators, government health departments, and pharmaceutical companies is required to ensure that pharmacogenetic drug labels reflect these recommendations and that they are readily accessible and actionable by healthcare professionals [10].

The Pre-emptive Pharmacogenomic Testing for Preventing Adverse Drug Reactions (PREPARE) study was initiated by the Ubiquitous Pharmacogenomics (U-PGx) consortium in 2017 to monitor the impact of pharmacogenomic testing prior to the dispensing of first prescriptions for 43 drugs/active pharmaceutical ingredients with actionable drug–gene pairs. Participants from hospitals, community centres, and pharmacies in seven European countries were genotyped for 50 germline variants in a 12-gene pharmacogenetic panel. Genotype guidance for prescriptions was revealed to significantly reduce the incidence of clinically relevant adverse drug reactions (ADRs), with a reduction of ~30% observed in cohorts in the United Kingdom [11].

African genomes harbour the greatest genetic diversity, with the African pan-genome revealing ~10% more DNA sequences than the GRCh38 reference human genome [12,13,14]. Furthermore, there has been a shift in the global research landscape to promote the inclusivity of diverse genomes to draft a more representative human pan-genome reference [15]. However, African populations have markedly distinct variations in genes involved in the absorption, distribution, metabolism, and excretion of drugs, with notable frequency differences in clinically relevant cytochrome P450 (CYP) variants. Not only are there differences in the allele frequencies compared with Caucasian and Asian populations, but intracontinental populations are also significantly distinct, with notable differences between South African and far West African populations [16,17]. Coupling this genetic diversity to the disease burden in Africa, the continent cannot be regarded a single entity when considering drug safety and efficacy, and there is a clear need for population-specific insights into pre-emptive pharmacogenetic screening [16,18].

Studies suggest that only 15% of drugs with clinical PGx guidelines have been comprehensively investigated in African populations, but uneven geographic distribution still limits the adequate representation of genetic diversity [19]. Knowledge gaps for African pharmacogenetics are increasingly being addressed by initiatives such as the Human Heredity and Health in Africa (H3Africa) consortium [17,20] and the African Pharmacogenetics consortium [21]. Furthermore, researchers are highlighting the vital importance of pharmacogenetic implementation and prioritization in treating hypertension [22], psychiatry [23], tissue rejection, pain management, cancer, tuberculosis, and others [24]. While the necessity for African-relevant PGx intervention towards improving equitable healthcare on the continent is clear, pre-emptive pharmacogenetics studies are yet to be implemented in South Africa (SA).

The aim of this study was to assess the dispensary trends for drugs which were monitored in the PREPARE study [11], derived from the DPWG actionable drug list, at a tertiary public hospital in SA. We highlight the differences in genetic diversity between European and African populations for the pharmacogenes monitored in the PREPARE study and stratify dispensary trends across drug classes, the number of prescriptions, and associated costs. Lastly, we utilize South African ADR reports to further frame the discussion on the potential impact which could be derived from the implementation of a PREPARE-like pharmacogenetics screening in SA.

## 2. Materials and Methods

### 2.1. Processing and Analysis of Hospital Dispensary Files

The Steve Biko Academic Hospital (SBAH) is a tertiary public healthcare institution in Pretoria (Gauteng province, South Africa) which serves as the main teaching hospital of the University of Pretoria. Dispensary data from January 2018 to November 2021 were obtained from the outpatient, inpatient, and oncology pharmacies at SBAH. These data comprised 12 separate data files, 1 for each dispensing pharmacy over the 4-year period. Each entry provided an item code number (ICN), emergency care network (ECN), drug name, drug dose (milligram, gram, millilitre, milligram/millilitre), drug formulation (capsule, table, cream, ointment, gel, solution, spray), route of administration (oral, injection, intra-muscular, sub-cutaneous, inhaled), number of units per prescription, total number of prescriptions dispensed, and total cost of prescriptions in South African Rand (ZAR). Where required, data were reorganized into individual columns and annotated with the appropriate year and pharmacy. Drugs from the DPWG actionable drug–gene interaction list which were monitored in the PREPARE study (*n* = 43), excluding oestrogen-containing drugs, were assessed, and referred to as SBAH-PGx. Due to low systemic absorption, prescriptions for topical indications (e.g., tacrolimus 0.1% ointment, fluorouracil 5% ointment) were excluded, although we acknowledge that patients with reduced DYPD gene activity may still experience adverse events from the topical application of fluorouracil [25]. The number of prescriptions per drug and cost per drug were stratified for each drug with the inclusion of dosage for each drug cumulatively, regardless of dosage, and per drug class.

### 2.2. Extraction of Allele Frequency Data

PGx gene-specific information tables created by PharmGKB and CPIC were accessed from the PharmGKB website (https://www.pharmgkb.org/page/pgxGeneRef accessed 8 September 2023). Frequency tables and allele functionality reference tables were accessed for *CYP2B6*, *CYP2C9*, *CYP2C19*, *CYP2D6*, *CYP3A5*, *DPYD*, *HLA-B*, *SLCO1B1*, *TPMT*, *UGT1A1*, and *VKORC1*. Oestrogen-containing drugs were excluded from the SBAH dataset in the analysis; the gene *fVl* was therefore excluded from allele frequency comparisons. Allele frequencies for European and sub-Saharan populations for each Single Nucleotide Polymorphism Database (dbSNP) Reference SNP cluster ID (rsID) number from the PREPARE study were compared. For genes where PharmGKB did not list frequency tables or allele functionality references (*CYP1A2*1C, CYP1A2*1F, UGT1A1*27*), allele frequencies for European (non-Finnish) and African American/African populations were obtained from the Genome Aggregation Database (gnomAD v3.1.2). *CYP2B6*16* has been merged as a sub-allele of *CYP2B6*18*; similarly, *SLCOB1*17* and *SLCOB1*15* were also merged and are therefore not reported separately. Additionally, due to the large number of functional gene duplications associated with *CYP2D6*, these were also not included in allele frequency comparisons.

### 2.3. Analysis of the South African Health Products Regulatory Authority VigiLyze Dataset

South Africa has been reporting ADRs since 1992 as a member of the World Health Organization (WHO) Programme for International Drug Monitoring (PIDM), and ranks number one for African countries in terms of individual case safety report (ICSR) submissions. The South African Health Products Regulatory Authority (SAHPRA) is the central body responsible for the reporting of ADRs to PIDM databases such as VigiBase—the World Health Organization’s global database for submission of ICSRs. ICSRs for 2018–2020 were obtained from SAHPRA. Files contained ICSR information as (1) cases with each Uppsala Monitoring Centre (UMC) report ID as a line item; (2) drugs each as a single line item; and (3) reactions each as a single line item. The list of drugs occurring in the SBAH-PGx dataset was filtered from the VigiLyze dataset with corresponding annotations as a concomitant, interacting, or suspected drug. For this analysis, ADR grading/seriousness criteria were not considered, but the number of reports associated with specific drugs were.

## 3. Results

### 3.1. Comparison of PREPARE Study Drugs with Steve Biko Academic Hospital Dispensary Data

During the review period (2018–2021), SBAH pharmacies dispensed a total of 6,798,880 prescriptions at a cumulative cost of ZAR ~369 million (Table 1). The outpatient pharmacy dispensed the most prescriptions (~63% of total prescriptions) at the highest total cost (50.5% of total expenditure), whereas the oncology pharmacy dispensed the fewest prescriptions (~11%) with the lowest total cost (17.5%). Of the 43 drugs monitored in the PREPARE study, 29 were dispensed from SBAH during the period spanning 2018–2021. Drugs which were not dispensed included acenocoumarol, atomoxetine, clomipramine, doxepine, escitalopram, flecainide, metoprolol, nortriptyline, oxycodone, phenprocoumon, pimozide, propafenone, and tegafur.

Of the 29 drugs, prescriptions associated with topical indications were excluded from further analyses. These included 440 prescriptions for fluorouracil ointment (total expenditure of ZAR 291,870.31) and 708 prescriptions for tacrolimus ointment (ZAR 263,082.23). These 29 drugs, termed the SBAH-PGx list, accounted for 695,256 prescriptions (~10% of total prescriptions) at a total cost of ZAR ~33 million (~9% of total expenditure) during the reporting period (Table 2 and Figure 1). Stratification by drug class revealed analgesics, cholesterol-lowering, and antidepressants (tri-cyclic antidepressants (TCA)) to be the most prescribed drug classes on the SBAH-PGx list, accounting for 76.7% of all prescriptions. In comparison, immunosuppressives, anticancer, and analgesics contributed to the highest cost burden, constituting 71.6% (ZAR 23,703,184.65) of the total cost. Collectively, these five drug classes, which included 14 drugs, accounted for 84.8% of the prescriptions and 84.1% of the cost attributed to the SBAH-PGx drug list. Table 2 includes the genes and number of dbSNPs which were monitored per class within the Euro-centric PREPARE study to provide a frame of reference for the complexity and scope of screening drug–gene pairs.

Several drugs had fewer than 100 prescriptions issued over the four-year period, including imipramine, codeine, zuclopenthixol, voriconazole, and paroxetine (Table 3). For codeine, only prescriptions for the single active ingredient were utilized for analysis, resulting in a low number of prescriptions in the SBAH-PGx list, which may have impacted the perceived prioritization of codeine in this study. Despite this, the analgesic tramadol had over 200,000 prescriptions in the SBAH-PGx-list, which contributed to the analgesics still being the most highly prescribed drug class (31%). Simvastatin (21%), amitriptyline (16%), atorvastatin (9%), and carbamazepine (4%) followed tramadol in the number of prescriptions dispensed. These five drugs, representing four different drug classes, constituted 81% of the prescriptions within the SBAH-PGx dataset. From a cost perspective, the top five individual drugs included tacrolimus (28%), tramadol (18%), capecitabine (13%), atorvastatin (7%), and mercaptopurine (5%), accounting for 71% of the cost within this dataset. Two drugs, tramadol and atorvastatin, were in the top five for both prescription number and cost; these are recommended to be monitored for 13 dbSNP variants for *CYP2D6* and 3 dbSNP variants for *SLCO1B1*, respectively. Additionally, data were interrogated with the inclusion of drug dosage and units dispensed per prescription and stratified based on the total number of units dispensed, as presented in Appendix A.

### 3.2. Population-Based Allele Frequency Comparisons

The SBAH-PGx list is based on the existing Euro-centric actionable drug–gene interactions which were prioritized within a specific study, and does not represent a comprehensive list of drug-gene interactions which would be the most impactful within SA. However, contextualizing these choices and the population-specific allele frequencies is critical to navigating a pathway to implementation. Allele frequencies for a total of 48 alleles from 12 genes were compared between populations of European and sub-Saharan African ancestry (Table 4). Population-based differences are evident in the allele frequencies for all 12 pharmacogenes, which is not surprising considering the genetic diversity within the African population. This table highlights instances where screening for specific alleles is essential, but also where certain alleles have low frequencies, which may not warrant screening in Africa. While this information is not novel, it could aid in prioritizing genetic variants for pre-emptive pharmacogenetic screening in Africa.

### 3.3. Analysis of the South African Health Products Regulatory Authority VigiLyze Dataset

Over the 3-year period, between 2018 and 2020, a total of 13,299 unique Uppsala Monitoring Centre (UMC) reports were filed from SA. Since all concomitant drugs are reported in conjunction with the suspected drug, the number of individually reported drugs was 37,739. When searching for the 29 drugs present in the SBAH-PGx drug list, 27 drugs were listed at least once as suspect, while 26 drugs were listed as concomitant, and 4 drugs listed as interacting, with thioguanine and mercaptopurine not having ICSRs for the period (Table 5 and Appendix A). Listings for efavirenz were separated by individual drug and fixed dose combinations (FDCs), thereby being represented as two separate line items in the dataset. These 27 drugs were suspected in a total of 2202 reports, which is ~16% of the total reports over a 3-year period. Notably, 12 of these 27 drugs recorded as suspected in fewer than 20 reports over the period. When stratified by class, anti-infectives (927), cholesterol-lowering drugs (421), and anticoagulants (214) were reported as having the most ADRs. With 906 reports, efavirenz was the most suspected drug in this subset of ADR reports. As a first-generation non-nucleoside reverse transcriptase inhibitor, efavirenz had been included in first-line therapy in SA since 2004. In 2019, the HIV clinical guidelines were revised to include a new formulation of FDC as first-line therapy, which includes the second-generation integrase inhibitor dolutegravir instead of efavirenz [26]. Therefore, efavirenz-associated ADR numbers are anticipated to decrease; however, these data support the case for the pre-emptive genotyping of *CYP2B6* in Africa to ensure that guidelines are proactive as opposed to reactive in response to potential ADRs. Atorvastatin, an HMGCoA reductase inhibitor indicated for hypercholesterolemia and individuals with high cardiovascular risk, had the third highest number of suspected reports, and was also in the top five drugs for highest cost and number of prescriptions. Simvastatin, the other cholesterol-lowering drug, was also in the top five of suspected ADRs, suggesting that the pre-emptive genotyping of *SLCO1B1* could also be beneficial in the South African health landscape.

## 4. Discussion

We have presented an analysis of regional hospital dispensary trends and national ADR reports framed around the potential benefits of implementing PGx in SA. These findings were contextualized around the recently published PREPARE pilot study, which reported a ~30% reduction in ADRs in response to pre-emptive PGx screening [11]. Although Euro-centric, the PREPARE-PGx framework was used comparatively to support the case for pre-emptive pharmacogenetic screening in SA by assessing national ADRs, prescription numbers, and associated costs for this actionable drug–gene list.

The discussion has been contextualized to support the case for prioritized regional pre-emptive PGx screening strategies in SA. Firstly, by highlighting several drugs identified as being widely dispensed or associated with high numbers of ADRs, and secondly, by underlining drugs which could be considered important within the context of SA but were not analysed within the PREPARE framework. Although comprehensive details of every actionable drug–gene pair are not discussed, we provide an informed perspective on pre-emptive PGx implementation.

As a tertiary hospital in SA, SBAH receives referrals from regional hospitals or local clinics which are not limited to provincial boundaries, making this hospital a suitable proxy for assessing regional dispensary trends. Cholesterol-lowering drugs [27], tri-cyclic antidepressants [28], opioid analgesics [29], and immunosuppressives [30], for which there are comprehensive genotype-based prescribing and dosing guidelines, constituted a significant proportion of total prescriptions and cost burden at SBAH. Furthermore, cholesterol-lowering drugs contributed to a significant portion of ADR reports alongside anti-infectives. Prescribing guidelines for cholesterol-lowering drugs (atorvastatin and simvastatin) are based on *SLCO1B1* genotyping to reduce statin-associated musculoskeletal symptoms. Few clinically relevant variants in *SLCO1B1* have functional impact, but decreased function reduces uptake in the liver, thereby increasing systemic exposure, resulting in ADRs [27].

Analysis of SA’s VigiBase data highlighted that low completeness scores hinder causality assessments, and consequently, result in an inability to inform signal detection [31,32]. This, combined with under-reporting and the uniqueness of ISCR report characteristics (products and types of ADR reports) between Africa and the rest of the world [33], creates a challenge for local prioritization. SA’s healthcare infrastructure is limited in its capacity to implement pre-emptive screening on par with the PREPARE study. It is therefore imperative to understand several factors before implementing pilot PGx studies in SA. These include local disease burdens, socioeconomic status, the number of prescriptions for drugs with clinical guidelines, and the type of ADRs experienced at a specific clinical setting (clinic, hospital, suburb, or province). While the ISCR reporting data presented here was not stratified by ADR-grading, which could be considered a limitation, this study provides an opportunity to reflect on the potential significance of several drug–gene pairs that could be considered for pre-emptive PGx pilot studies locally.

The analgesic tramadol was the most frequently prescribed drug from the SBAH-PGx list, and its actionable gene pair is the highly polymorphic *CYP2D6*. The high frequency of several decreased function alleles in sub-Saharan African population suggests that ineffective pain management could be prevalent in clinical healthcare. However, due to the magnitude of prescriptions dispensed, the potential for actioning pre-emptive pharmacogenetic screening in a country not yet equipped for implementation seems insurmountable. In contrast, the anti-cancer drug tamoxifen had 10-fold fewer prescriptions; thus, screening *CYP2D6* for high-frequency alleles (*29, *17, *10, *41) would result in tangible benefits for patient’s clinical outcomes.

Tacrolimus, the mainstay immunosuppressive used post solid organ transplant, is another example to reflect on, as it accounted for 1.5% of SBAH-PGx prescriptions but 28% of the cost. While therapeutic drug monitoring is implemented, genotyping before initiation has implications for achieving therapeutic concentrations post transplant. *CYP3A5* is the actionable drug–gene pair and the *3 allele, present in 92% of the European population, is the basis on which tacrolimus dosage is initiated. African populations, however, predominantly harbour a *1 allele [30]. Genotyping for *1, *3, *6, and *7 alleles prior to organ transplant could improve patient outcomes and the management of life-long immunosuppressant regimens.

Warfarin, which was associated with the third highest number of ADR reports in this dataset, is also one of the most prescribed oral anticoagulants in sub-Saharan Africa. African patients exhibit sub-optimal therapeutic ranges for warfarin, and genetic variants within several genes [34] have been shown to impact warfarin dose variability in these populations [35].

One of the most well studied drug–gene interactions which impacts the African continent is that of *CYP2B6* and efavirenz. With a higher frequency of reduced function (*6) and non-functional (*18) alleles in sub-Saharan African populations, overlayed with the significant burden of HIV, the value of PGx has been evident for almost a decade. However, despite understanding the consequences of efavirenz toxicity due to absence of genotype information [36], SA has been reactive to the growing burden of significant ADRs by removing efavirenz as the recommended first-line therapy [37]. Healthcare systems in SA still rely on reactive instead of pro-active approaches, and the role of polypharmacy and changes in local recommendations require consideration.

This analysis, however, highlights the need for Afro-centric considerations for drug–gene pairs, which would require monitoring in SA for drugs used in tuberculosis (TB) and malaria. SA is listed as a high burden country for TB, HIV-associated TB, and multi-drug resistant/rifampicin-resistant TB [38]. Slow-acetylator genotypes are prevalent in SA [39] and can contribute to drug-induced hepatotoxicity [40]. Isoniazid was implicated as the suspect drug in 192 ADR reports and associated as concomitant in 351 additional reports (data not shown); although CPIC guidelines are not yet published for *NAT2*, as of 2023, the burden of TB may justify consideration in an African PGx setting. Similarly, dose-dependent haemolysis linked with *G6PD* deficiency is associated with anti-malarial drugs, such as primaquine, which results in considerable uncertainty in policies and practices [41]. With regional relevance in sub-Saharan Africa coupled to the variable distribution of *G6PD* polymorphic variation, the opportunity to locally implement guidelines in the context of *G6PD* genotyping would improve treatment outcomes and reduce ADRs [42].

The data presented here further support previous efforts calling for prioritization within disease areas and/or drug–gene pairs or drug classes within Southern Africa [43,44,45]. With the clinical genotyping of South African and Zimbabwean cohorts using a 46 gene panel revealing that 100% of participants harboured at least one actionable PGx variant, with a median distribution of four actionable variants [46], foregoing implementation has serious consequences for patients in sub-Saharan Africa. While concerns have been raised regarding the magnitude of ADR reduction reported in the PREPARE study, along with the limitations of focusing on only ADRs as an outcome in pharmacogenetic testing [47,48], it has also been argued that the reported effect size remains plausible due the limited scope and scale of the study. Nonetheless, these findings still present an intriguing prospect in terms of the potential benefits that could be realized from a similar PGx pilot in a population group with greater genetic diversity.

Despite regular updates to guidelines, incorporating pharmacogenomics into routine clinical practice has seen a slow uptake in implementation [49,50]. Attitude and perceptions studies suggest that (i) inadequate knowledge on how to utilize guidelines in combination with clinical results, (ii) competency required in pharmacogenomic counselling, (iii) lack of awareness, and (iv) concerns over costs have impacted healthcare practitioners and patients [51]. Several knowledge, attitude, and perceptions studies in Africa have highlighted similar concerns [52,53,54]. Africa still faces several overarching issues which impact PGx, including (i) a general lack of representative genomics studies; (ii) the scarcity of functional studies on genetic associations with drug responses in African populations; (iii) clarity regarding ethnic classifications within existing data; (iv) the relevance of Euro-centric clinical pharmacogenetics tests to African populations; (v) historical data gaps for pharmacokinetic studies of older drugs; and (vi) fiscal challenges associated with the education of healthcare professionals and scientists in resource-limited settings [19].

Despite these seemingly overwhelming hurdles, here we make a case for starting PGx implementation in SA without demanding extensive multi-gene panels in every healthcare facility across the country. A practical approach for evaluating the potential benefits of PGx locally could focus on harnessing contextual clinic-based expertise as the driving force behind initial PGx implementation; for example, based on the data presented in this study, an oncology centre might realize greater impact by focusing on relevant *CYP2D6* alleles for PGx, while a specialist organ transplant centre would benefit from a *CYP3A5*-centred PGx strategy.

## 5. Conclusions

African populations face a unique burden of disease which is further compounded by diverse genetics, unique environmental factors, socioeconomics, and fiscal constraints. The interactions between these factors across geospatial borders ultimately define the state of healthcare for various population groups across the African continent. Despite being hampered by these same challenges, intentional and comprehensive investigations into regional or clinic-specific dispensing patterns across South Africa, for drug–gene pairs which have CPIC guidelines, could inform and prioritize regional pre-emptive PGx screening strategies. Current investments in pre-emptive PGx screening may realize significant long-term benefits through improved clinical outcomes and fiscal relief from expenditures related to ADRs. The people of SA, governments, regulators, and healthcare providers would therefore all stand to benefit from the implementation of PGx pilot studies within the country. While large-scale national screening efforts should ultimately guide the long-term vision of a comprehensive African PGx program, regional genomic diversity, disease burden, and first-line treatment options could be harnessed to inform stratified PGx today.

## Figures and Tables

**Figure 1 jpm-14-00114-f001:**
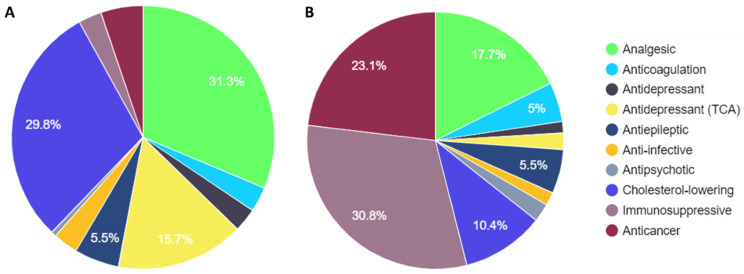
Graphical representation of the percentage contribution per drug class for (**A**) the number of prescriptions and (**B**) expenditure at SBAH for the SBAH-PGx drug list.

**Table 1 jpm-14-00114-t001:** Prescription numbers and associated cost for each pharmacy per year between 2018 and 2021.

Pharmacy	Year	Prescription Number	Cumulative Prescriptions	Total Cost (ZAR)	Cumulative Cost (ZAR)
Inpatient	2018	477,254	1,751,386	28,900,337.69	117,892,639.02
	2019	459,999		30,319,287.36	
	2020	453,137		31,785,757.32	
	2021	360,996		26,887,256.65	
Outpatient	2018	1,169,679	4,288,797	43,350,476.40	186,686,832.97
	2019	1,106,479		45,458,498.01	
	2020	1,103,182		52,555,138.32	
	2021	909,457		45,322,720.24	
Oncology	2018	205,341	758,697	18,034,013.16	64,540,893.00
	2019	207,669		13,907,246.91	
	2020	198,983		24,021,552.54	
	2021	146,704		8,578,080.39	
Total	2018–2021	6,798,880		369,120,364.99	

**Table 2 jpm-14-00114-t002:** Total number of prescriptions at Steve Biko Academic Hospital and cost for each drug class over the period of 2018–2021, linked to the number of genes and dbSNPs monitored per class in the PREPARE study.

Drug Class	Prescription Number	Total Cost (ZAR)	Genes Monitored	dbSNPs Monitored
Analgesic	217,572	5,848,133.71	*CYP2D6*	13
Cholesterol-lowering	207,018	3,444,997.81	*SLCO1B1*	3
Antidepressant (TCA)	109,345	684,720.60	*CYP2D6*/*CYP2C19*	23
Antiepileptic	38,216	1,834,534.52	*HLA B*5701*/*CYP2C19*	6
Anticancer	37,801	7,648,288.64	*DPYD*/*UGT1A1*/*CYP2D6*	20
Anticoagulation	21,370	1,642,319.46	*CYP2C9*/*CYP2C19*/*VKOR1*	11
Anti-infective	21,357	553,183.30	*CYP2B6*/*HLA B*5701*/*CYP2C9*	14
Antidepressant	20,180	476,304.87	*CYP2D6*/*CYP2C19*	23
Immunosuppressive	18,123	10,206,762.30	*TPMT*/*CYP3A5*	6
Antipsychotic	4274	762,684.76	*CYP2D6*/*CYP1A2*	15
Total	695,256	33,101,929.97		

**Table 3 jpm-14-00114-t003:** Each drug represented in the SBAH-PGx list was stratified according to total number of prescriptions issued between 2018 and 2021 with its associated cost.

Drug Class	Description	Total Number of Prescriptions Issued	Cost (ZAR)
Analgesic	Tramadol	217,546	5,836,578.7
Cholesterol-lowering	Simvastatin	142,450	1,205,186.0
Antidepressant (TCA)	Amitriptyline	109,340	684,461.7
Cholesterol-lowering	Atorvastatin	64,568	2,239,811.8
Antipileptic	Carbamazepine	29,287	1,307,022.6
Anticancer	Tamoxifen	20,048	765,132.7
Anti-infective	Flucloxacillin	17,754	241,172.0
Antidepressant	Citalopram	15,969	176,589.1
Anticoagulant	Clopidogrel	11 157	910,518.3
Anticancer	Fluorouracil	10,971	281,973.2
Anticoagulant	Warfarin	10,213	731,801.2
Immunosuppressive	Tacrolimus	10,156	9,413,587.5
Antiepileptic	Phenytoin	8929	527,511.9
Immunosuppressive	Azathioprine	7967	793,174.8
Anticancer	Capecitabine	4238	4,386,219.7
Anti-infective	Efavirenz	3565	174,476.0
Antipsychotic	Haloperidol	2965	88,551.0
Antidepressant	Sertraline	2108	156,890.4
Antidepressant	Venlafaxine	2059	138,826.1
Immunosuppressive	Mercaptopurine	1508	1,600,266.0
Antipsychotic	Aripiprazole	1032	648,508.4
Anticancer	Irinotecan	857	213,280.9
Antipsychotic	Clozapine	247	23,949.7
Immunosuppressive	Thioguanine	179	401,416.1
Antidepressant	Paroxetine	44	3999.3
Anti-infective	Voriconazole	38	137,535.3
Antipsychotic	Zuclopenthixol	30	1675.7
Analgesic	Codeine	26	11,555.0
Antidepressant (TCA)	Imipramine	5	258.9

**Table 4 jpm-14-00114-t004:** Comparison of allele frequencies between populations of European and sub-Saharan African ancestry with the fold enrichment in allele frequency between the populations.

Genes	Allele	dbSNP rsID Number	Functional Status	Allele Frequency European	Allele Frequency Sub-Saharan Africa	Fold Enrichment
*CYP3A5*	*6	rs10264272	No function	0.00151	0.19324	127.6
*CYP2D6*	*29	rs61736512/ rs59421388	Decreased function	0.00105	0.10833	103.5
*CYP2C9*	*5	rs28371686	Decreased function	0.00017	0.01033	59.2
*UGT1A1*	*37	rs8175347	Decreased function	0.00069	0.03707	54.0
*CYP2D6*	*17	rs28371706	Decreased function	0.00392	0.19355	49.4
*CYP2C9*	*8	rs7900194	Decreased function	0.00181	0.07585	42.0
*CYP2C19*	*9	rs17884712	Decreased function	0.00066	0.02696	40.8
*CYP1A2* ^	*1C	rs2069514	Decreased function	0.01298	0.27690	21.3
*CYP2C9*	*11	rs28371685	Decreased function	0.00164	0.02569	15.6
*TPMT*	*3C	rs1142345	No function	0.00492	0.05288	10.7
*UGT1A1* ^	*27	rs35350960	Decreased function	0.00001	0.00004	5.2
*CYP2D6*	*10	rs1065852	Decreased function	0.01571	0.04869	3.1
*CYP2D6*	*5	Deletion	No function	0.02948	0.06209	2.1
*CYP2C19*	*3	rs4986893	No function	0.00162	0.00267	1.6
*CYP2B6*	*6	rs3745274	Decreased function	0.23298	0.37487	1.6
*UGT1A1*	*28	rs8175347	Decreased function	0.31647	0.40004	1.3
*CYP2C19*	*2	rs4244285	No function	0.14686	0.15684	1.1
*CYP1A2* ^	*1F	rs762551	Increased function	0.71180	0.60500	0.8
*CYP2C19*	*17	rs12248560	Increased function	0.21544	0.17334	0.8
*CYP2D6*	*41	rs28371725	Decreased function	0.09238	0.04529	0.5
*VKORC1*	X	rs9934438	Decreased function	0.41326	0.10774	0.3
*CYP3A5*	*3	rs776746	No function	0.92438	0.24095	0.3
*SLCO1B1*	*15	rs4149056	No function	0.15017	0.02793	0.2
*HLA-B*5701*	X	rs2395029		0.03604	0.00609	0.2
*CYP2D6*	*4	rs3892097	No function	0.18485	0.02873	0.2
*CYP2C9*	*3	rs1057910(C)	No function	0.07554	0.01116	0.1
*CYP2C9*	*2	rs1799853	Decreased function	0.12730	0.01311	0.1
*CYP3A5*	*7	rs41303343	No function	0.00000	0.08641	n/a
*CYP2D6*	*3	rs35742686	No function	0.01592	0.00098	n/a
*CYP2B6*	*18	rs28399499	No function	0.00000	0.05768	n/a
*CYP2C19*	*4A/B	rs28399504	No function	0.00236	0.00000	n/a
*CYP2C19*	*5	rs56337013	No function	0.00003	0.00000	n/a
*CYP2C19*	*6	rs72552267	No function	0.00030	0.00000	n/a
*CYP2C19*	*7	rs72558186	No function	0.00000	0.00000	n/a
*CYP2C19*	*8	rs41291556	No function	0.00336	0.00000	n/a
*CYP2C19*	*10	rs6413438	Decreased function	0.00000	0.00000	n/a
*CYP2D6*	*6	rs5030655	No function	0.01120	0.00000	n/a
*CYP2D6*	*8	rs5030865	No function	0.00022	0.00000	n/a
*CYP2D6*	*9	rs5030656	Decreased function	0.02754	0.00000	n/a
*CYP2D6*	*14A/B	rs5030865	Decreased function	0.00000	0.00000	n/a
*DPYD*	*2A	rs3918290	No function	0.00792	0.00000	n/a
*DPYD*	*13	rs55886062	No function	0.00056	0.00000	n/a
*DPYD*	c.2846 A>T	rs67376798	Decreased function	0.00374	0.00000	n/a
*DPYD*	c.1236G>A	rs56038477	Decreased function	0.02374	0.00000	n/a
*SLCO1B1*	*5	rs4149056	No function	0.02040	0.00000	n/a
*TPMT*	*2	rs1800462	No function	0.00206	0.00000	n/a
*TPMT*	*3B	rs1800460	No function	0.00283	0.00000	n/a
*UGT1A1*	*6	rs4148323	Decreased function	0.00787	0.00000	n/a

^ allele frequencies for *CYP1A2*1C*/*CYP1A2*1F*/*UGT1A1*27* are from gNOMAD v3.1.2 for individuals of European (non-Finnish) and African American/African ancestry. If allele frequencies within either population were zero—fold enrichment was annotated as not applicable (n/a).

**Table 5 jpm-14-00114-t005:** Number of ADR reports per drug associated as concomitant, interacting, or suspected which were associated with the SBAH-PGx list.

Drug Class	Drug	Concomitant	Interacting	Suspected
Anti-infective	Efavirenz FDC	71	1	522
Anti-infective	Efavirenz	179	1	384
Cholesterol-lowering	Atorvastatin	260		285
Anticoagulant	Warfarin	78	1	178
Cholesterol-lowering	Simvastatin	313		136
Antidepressant	Citalopram	62		125
Antidepressant (TCA)	Amitriptyline	89		95
Analgesic	Tramadol	59		84
Antiepileptic	Carbamazepine	36		64
Antipsychotic	Clozapine	4		59
Antipsychotic	Haloperidol	15		48
Anticoagulant	Clopidogrel	52		36
Antiepileptic	Phenytoin	14		36
Antidepressant	Sertraline	26		24
Antipsychotic	Zuclopenthixol	14		22
Immunosuppressive	Tacrolimus	9		21
Anticancer	Capecitabine	3		18
Anti-infective	Voriconazole	0		17
Immunosuppressive	Azathioprine	16		13
Antipsychotic	Aripiprazole	5		11
Analgesic	Codeine	15		6
Anticancer	Tamoxifen	1		5
Anti-infective	Flucloxacillin	1		4
Anticancer	Irinotecan	1		3
Antidepressant	Paroxetine	8		3
Anticancer	Fluorouracil	2		2
Antidepressant (TCA)	Imipramine	1		1
Antidepressant	Venlafaxine	20	4	0
Total		1354	7	2202

## Data Availability

Raw or processed data files are available on request from the corresponding author, in accordance with REC requirements.

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
