# Peer review of "The Case for Pre-Emptive Pharmacogenetic Screening in South Africa"

_jpm, 2024, doi:10.3390/jpm14010114_

Round 1

Reviewer 1 Report

Comments and Suggestions for Authors

The authors state the following as a primary concern and motivation for their research project:

Lack of equitable representation of global genetic diversity has hampered the implementation of genomic medicine in underrepresented populations including those on the African continent.”

 Importantly, lack equitable representation is prominent and broadly acknowledge to be a deficit and weakness of the GWAS data sets.

Although many westernized countries have government funded national genomic medicine programs, the African continent is lacking such a program.

Barriers to implementing preemptive pharmacogenomics in African continent: 

a) Historically, lack of evidence based guidelines was a barrier to developing national phramcogenomic programs.  But now, with professional societies and initiative such as CPIC, DPWG, PharmGKB, continualy curated and annoted evidence based guidelines are available and applicable for the population in many westernized countries.

2) Importantly, but also adding to complexity is that fact that African genomes have more diversity that other genome populations that have been studied thus far, such as Caucasian and Asian.  There is also significant diversity within different populations within the African continent, adding the complexity, with diversity in cytochrome P450 enzymes being a prominent  factor.

The authors compare Europeans and African populations (through data available in medical records at Steve Biko Acadamic Hospital in South Africa) in terms of pharmacogenomic genetic diversity (frequency of variant allele functionality) , dispensing trends of specific drugs, costs associated with use of specific drug classes, reported adverse drug reactions.

The authors found that 27 of 29 drugs that were previously identified in the PREPARE study (conducted in 7 European countries) and which accounted for 10% of total expenditures in the PREPARE study and 9% of total expenditures in the South African (SA) study.

Results found that allele frequency population differences for all 12 pharmacogenes evaluated

Regional relevance of pharmacogenomic variance impact is exemplified in diseases such as HIV, tuberculosis and malaria.

The authors conclude that their research results support calling for greater efforts towards “prioritization of drug-gene pairs or drug classes within South Africa” and suggest ways of moving towards this goal that might initially involve taking smaller steps, to make the process more feasible while moving steadily towards more comprehensive long term goals, such as initially testing smaller gene panels that are known to play a significant role in pharmacogenomics outcomes in the African population.

This is an excellent article. It is highly relevant and highlights the critical unmet needs involving significant disparity of pharmacogenomics knowledge and applicability of meaning pharmacogenomics information when comparing European populations to African populations.

Notably, the authors did not provide data related to drug-gene interaction risks for malignant hyperthermia (e.g., the CACNA1S gene and RYR1 gene). As variants in such genes have potential for acute adverse reactions that are often not reversible and lead to rapid mortality in many cases, it would be important to understand the role and impact that preemptive testing of such genes in the African population would cause.  It would be helpful it the authors commented on the drug-gene interactions that lead to malignant hyperthermia and why this data was not examined for this study. A reference to this topic includes: Raga SV, Wilmshurst JM, Smuts I, et al. A case for genomic medicine in South African paediatric patients with neuromuscular disease. Front Pediatr. 2022;10:1033299. Published 2022 Nov 17. doi:10.3389/fped.2022.1033299

Topical products such as  5-FU (fluorouracil) cream and topical tacrolimus were excluded from the study.  Topical 5-FU has minimal systemic absorption, but must still be metabolized by DYPD gene would be involved in metabolism of tropical 5-FU. Theoretically, patients with reduced DYPD gene activity may experience increased adverse events from topical 5-FU. (Membrive Jiménez C, Pérez Ramírez C, Sánchez Martín A, et al. Clinical Application of Pharmacogenetic Markers in the Treatment of Dermatologic Pathologies. Pharmaceuticals (Basel). 2021;14(9):905. Published 2021 Sep 6. doi:10.3390/ph14090905).  Do the authors want to comment on this? 

Author Response

Please see the attachment,

Reviewer 2 Report

Comments and Suggestions for Authors

This study describes the potential impact of pre-emptive pharmacogenomics screening in South Africa. The manuscript is clear, relevant for this specific field, and presented in a well-structured manner. The results provide an advancement of the current knowledge, especially for the African continent, to support the benefit of pre-emptive pharmacogenomics screening in South Africa to promote and initiate its application. However, the limits to the application of these tests are also general: the existence of discrepancies in terms of indications between the pharmacogenetic guidelines produced by the different organizations/regulatory entities; the lack in some cases of robust evidence demonstrating the cost-effectiveness of pharmacogenomic analysis; the absence of support for the clinician to choose the right test to the right patient or interpret the result of the same in order of the right suggested dose of a drug.

However, I find the data reported and the conclusions consistent.

Some inaccuracies need to be corrected in the text before publication: gene names should be written in capital italics; in Table 2 correct the name of the reported genes DPYD/UGT1A1 and TPMT; the order in Table 4 could be alphabetical to facilitate reading and SLC01B1 would be replaced by SLCO1B1 for two times; in line 349 is G6PD deficiency.

Comments on the Quality of English Language

The English language is appropriate and understandable.

Reviewer 3 Report

Comments and Suggestions for Authors

This study presents an analysis of trends in regional hospital dispensaries and national reports in South Africa (SA) on ADRs, focusing on the potential benefits of implementing PGx in SA. The results were contextualized based on the PREPARE pilot study which reported an approximately 30% reduction in ADRs in response to preventive PGx screening. Although Eurocentric, the PREPARE-PGx framework has been used comparatively to support the case for preventive pharmacogenetic screening in SA by assessing national ADRs, number of prescriptions, and associated costs for a list of actionable drug-genes.

I have the following concerns:

Abstract

-        Page 1, line 21

Insert the abbreviation (SA) after South Africa, as you then use it in line 28.

-        Page 1, line 26

“Furthermore, a comparison of allele frequencies of pharmacogences between African and European populations were used to frame the potential impact of pre-emptive pharmacogenomics screening in SA.”

Replace the word “pharmacogences” with “pharmacogenes”.

-        Page 1, line 28

I suggest briefly reporting the drugs (e.g., tamoxifen, tacrolimus, atorvastatin, efavirenz) and their respective alleles for which pharmacogenetic screening in SA seems essential, based on prescriptions, costs, and ADRs.

-        Page 1, lines 28-30

“Enumerating the benefit of pre-emptive pharmacogenomics screening in SA will only be possible once we initiate its application. However, regional genomic diversity, disease burden, and first-line treatment options could be harnessed to inform stratified PGx today.”

Replace with:

Enumerating the benefit of pre-emptive pharmacogenomics screening in SA will only be possible once we initiate its full application. However, regional genomic diversity, disease burden, and first-line treatment options could be harnessed to target stratified PGx today.

Introduction

-        Page 2, lines 46-48

“Notably, to date, there are no active government-funded national genomic medicine programs underway on the African continent, thereby limiting the realization of the benefits of genomic healthcare for the continent [3].”

Replace the word “Notably” with “In contrast”.

-        Page 2, lines 61-63

“While each committee’s guidelines are nuanced, their evidence-based recommendations support clinical decision-making [9].”

Insert the comma in bold.

Materials and Methods

-        Page 4, lines 156-160

“The list of drugs occurring in the SBAH-PGx dataset were filtered from the VigiLyze dataset with their corresponding annotation as concomitant, interacting, or suspected drug. For this analysis, ADR grading/ seriousness criteria were not considered, but the number of reports associated with specific drugs were.”

I suggest that the lack of grading/seriousness criteria for ADRs, understood as a limitation of the study, should also be brought back into the discussion.

Results

Page 7, lines 234-236

“The SBAH-PGx list is based on the existing Euro-centric actionable drug-gene interactions which were prioritized within a specific study and does not represent a comprehensive list which would be the most impactful within SA”

I think it’s more correct to put this concept, intended as a limitation of the study, in the discussion section.

Page 8, lines 241-245

“This table highlights instances where screening for specific alleles is essential but also where certain alleles have low frequencies which may not warrant screening in Africa. While this information is not novel, it could aid in prioritizing genetic variants for preemptive pharmacovigilance screening in Africa.”

I recommend highlighting with some examples those alleles with high frequencies for which screening is essential in Africa (e.g. CYP2D6-*17, -*10, CYP2B6-*6) and those with low frequencies (DPYD-*2A, -*13) that, therefore, do not warrant screening in Africa.

Replace the word “pharmacovigilance” with “pharmacogenetics”.

Page 9, lines 264-266

“When stratified by class, antiinfectives (927), cholesterol-lowering drugs (421), and anticoagulants (214) were reported as having the most ADRs.”

Considering the extensive use of anticoagulants drugs, pharmacogenetics has particular importance also in this field. Several polymorphisms influence the response to anticoagulant agents and tests, based on their identification, are now available. However, despite accumulating evidence on the utility and feasibility of some pharmacogenetics tests, several barriers still exist in their implementation in clinical practice. Based on your reports, anticoagulants, particularly warfarin, also cause many ADRs in Africa (Table 5). I recommend devoting a few lines of the discussion to the African situation in this area.

Conti V, Manzo V, De Bellis E, Stefanelli B, Sellitto C, Bertini N, Corbi G, Ferrara N, Filippelli A. Opposite Response to Vitamin K Antagonists: A Report of Two Cases and Systematic Review of Literature. J Pers Med. 2022 Sep 25;12(10):1578. doi: 10.3390/jpm12101578.

Dahal, K.; Sharma, S.P.; Fung, E.; Lee, J.; Moore, J.H.; Unterborn, J.N.; Williams, S.M. Meta-Analysis of Randomized Controlled Trials of Genotype-Guided vs Standard Dosing of Warfarin. Chest 2015, 148, 701–710

Page 9, lines 267-270

“As a first-generation non-nucleoside reverse transcriptase inhibitor, efavirenz was included in first-line therapy in SA from 2004. In 2019, the HIV clinical guidelines were revised to include a new formulation of FDC as first-line therapy which includes the second -generation integrase inhibitor dolutegravir instead of efavirenz.”

Insert references.

Discussion

-        Page 11, lines 326-328

“CYP3A5 is the actionable gene-pair and the *3 allele, present in 92% of the European population, is the basis on which tacrolimus dosage is initiated.”

What is meant by gene-pair? Perhaps the drug-gene pair is meant. In this case, improve sentence construction.

-        Page 12, line 340

Given the numerous ADRs reported during treatment with hypocholesterolemic agents such as atorvastatin and simvastatin, it is worth discussing the importance of pharmacogenetic screening of SLCOB1.

-        Page 12, lines 354-355

“The data presented here further supports previous efforts calling for the prioritization of drug-gene pairs or drug classes within Southern Africa [39].”

Better organize the speech. Specify that previous efforts were pushing in prioritizing pharmacogenetic screening for certain drugs and thus of certain drug-gene pairs.

Tables

Note that the abstract, main text, and figure/table/scheme captions are treated separately for abbreviations. This means that you need to define the abbreviation the first time you use it in each part—you may have to define the same abbreviation three separate times. The reason for this is that they are often displayed in isolation; for example, indexing services usually only display the abstract and you can browse figures without the main text via the journal website.

-        Table 2

“Total number of prescriptions and cost for each drug class over the period of 2018-2021, linked to the number of genes and dbSNPs monitored per class in the PREPARE study.”

Replace with:

Total number of prescriptions at Steve Biko Academic Hospital (SBAH) and cost for each drug class over the period 2018-2021, linked to genes and dbSNPs monitored by class in the PREPARE study."

The total number of prescriptions (695 256) reported in Table 2 does not match with the total number of prescriptions written in the text (page 5, line 179).

References

Formatting references using the ACS reference style.

Example:

Fisher, J.A.; Krapf, C.B.E.; Lang, S.C.; Nichols, G.J.; Payenberg, T.H.D. Sedimentology and architecture of the Douglas Creek terminal splay, Lake Eyre, central Australia. Sedimentology 2008, 55, 1915–1930.

Comments on the Quality of English Language

Minor editing of English language required
